# Classroom Psychomotor Education Programme to Enhance Executive Functions: A Cluster Randomised Feasibility Trial

**Vassiliki Riga *** and **Aimilia Rouvali**

Department of Educational Sciences and Early Childhood Education, University of Patras,
26504 Rio Achaia, Greece; erouvali@yahoo.gr
* Correspondence: vriga@upatras.gr

**Abstract:** An increase in children exhibiting attention difficulties has created the need for more classroom-based intervention programmes. A promising link between physical education and improvement in executive functioning has been at the centre of attention. POTENTIAL constitutes a novel classroom-based psychomotor education programme to enhance students' attention and listening skills by improving executive functions. A cluster randomised feasibility study was conducted to explore the feasibility of a definitive trial to assess POTENTIAL's effectiveness regarding (i) recruitment and sampling procedures; (ii) compliance and fidelity; (iii) the acceptability of POTENTIAL by teachers and children; and (iv) the appropriateness of the outcome measures. Four early years classes with an inclusion unit participated: two implemented POTENTIAL and two received no intervention. Eight children in each class (*n* = 32) were sampled to investigate the appropriateness of the outcome measures. Teachers' diaries were utilised to explore the acceptability of the activities and the fidelity and compliance to the implementation. The findings regarding POTENTIAL's acceptability were positive. The recruitment targets were met, and compliance and fidelity were good. Mixed results were produced about the appropriateness of the outcome measures. Thus, the trial protocol could be scaled up in a definitive trial. This study highlights the need for more physical education programmes to support children's executive functioning.

**Keywords:** preschool education; attention and listening skills; psychomotricity; executive functioning; working memory; inhibition control; cognitive flexibility

## 1. Introduction

Every child's first years constitute a critical developmental stage [1,2]. During this period, the brain and central nervous system evolve rapidly as new synapses between cells are formed [3]. This period is also characterised by the instantaneous development of two overlapping and interrelated areas; motor and cognitive [4]. Lubans and his colleagues [5] highlighted a few mechanisms (neurobiological, psychosocial, and behavioural) that link Physical Activity (PA) and cognition. Further research in the field indicates that engagement in PA results in brain alterations (e.g., increased cerebral blood flow and oxygenation, neurogenesis) [6], changes in mood and emotions, and self-regulation skills [7,8]. Additionally, PA interventions have been found beneficial to children's cognitive functioning [9], contributing to "the mental acuity, skills, and strategies that are important for navigating challenges faced across the life span" ([10], p. 1198). It comes as no surprise that countries such as Australia [11] and Canada [12] have introduced new guidelines suggesting that children between five to eleven years of age should engage in PA daily for at least 60 min [11] as a means of enhancing and supporting their overall development. The significance of PA becomes even more crucial, considering that PA habits formed and established during early childhood have been tracked into middle childhood and even adulthood [13]. Thus, should we aim for life-long overall health and well-being, more emphasis should be given to promoting physical literacy in the early years. Especially regarding early years, PA

and PA programmes have been positively linked to several health outcomes [14], school readiness, and academic success [15].

However, over the past decade, the amount of PA that early years children engage in their everyday lives has decreased significantly. A key factor behind this new reality, according to Corcoran and Steinley [16], is that more children spend most of their time in educational settings where, nowadays, the emphasis is given to standardised performance at the expense of PA and other more age-appropriate opportunities and activities (like play). Fortunately, classroom-based PA interventions have gained much attention, with a significant number of researchers focusing on developing them to support young children's overall development (motor, cognitive, and psychological) [17,18] while promoting the idea of physical literacy, a concept that has gained momentum globally over the last few years [19]. This study aims to add to this significant matter innovatively.

### 1.1. Rationale for Developing an Intervention for Executive Functions

Lately, research has pinpointed a steady increase in the rate of children exhibiting attention and listening difficulties in the classroom [20]. There are many reasons a child may experience attention difficulties in preschool: language disorders, Attention Deficit Hyperactivity Disorder (ADHD), hearing loss or difficulties, or forms of psychopathology. Attentional difficulties have also been associated with children's low socio-economic backgrounds [21]. However, it can also be a part of typical child development. It is evident then that those attention-related difficulties (especially during preschool) are shared among all children. Research has shown a direct association of attention problems in preschool with later problems in reading, writing, and mathematics, as well as difficulties in children's social interactions and emotional development [22–24]. A common characteristic among all children in the above groups is the difficulties or deficits in their executive functions (EFs). Whether they exhibit difficulty keeping and manipulating information, holding back reactions, or following commands, children with attention deficits also exhibit EF difficulties and vice versa.

A dichotomy has characterised the research in the field of EFs regarding the existence of precise terminology for them. However, it has been accepted that EFs do not constitute a unitary process. Instead, it is an umbrella term often used to describe the higher cognitive functions responsible for an individual's ability to act purposefully, goal-directed, adaptively, and flexibly in a challenging environment characterised by novel or complex situations [7,25]. Unlike other aspects of cognition considered fixed, EFs are more fluid [26]. EFs consist primarily of three distinct but interrelated subfunctions: working memory, inhibitory function (also inhibitory control or inhibition), and cognitive flexibility (or shifting), which start developing during infancy and continue to develop into adulthood, with infancy being one of the most critical stages [27]. Due to the existence of various definitions for each EF, in the context of the current study, the definition used for each of them is presented below.

Working memory is a person's ability to retain, monitor, and manipulate information for a short period. Working memory is divided into verbal and visuospatial, depending on the form of information that the person is asked to process [25,27].

Inhibitory refers to the individual's ability to overcome dominant responses and conflicting stimuli. The inhibition process consists of two separate factors in the existing literature under various names. More specifically, inhibition consists of response inhibition (also referred to as 'behavioural inhibition', 'motor inhibition', 'dominant response inhibition', and 'attentional restraint') and attentional inhibition (also referred to as 'control interference', 'interference suppression', and 'resistance to interference') [25,28–33]. In the context of the present study, the names proposed by Diamond [25] are used, which are 'behavioural inhibition' and 'attentional inhibition/selective-sustained attention'.

Cognitive flexibility refers to the individual's ability to mentally shift between tasks, rules, and mental sets to develop appropriate behaviours and responses [34].

Attention is the behavioural and cognitive process in which the individual remains selectively focused on a particular stimulus or set of stimuli while ignoring other perceived information. According to Posner's model of attention, the ability to pay attention is linked to the functioning of three different neural systems that aim at: (1) alertness, (2) attentional orientation, and (3) regulation of thoughts, feelings, and actions [35].

In the context of the present study, the term 'self-regulation' is also used to refer to the behavioural aspects of self-regulation (e.g., working memory, inhibitory control, and attention), which are essential for the planning and execution of targeted actions [36].

Recently, there has been increasing interest in EFs relating to children's social and emotional development and academic success [37]. EFs, in addition to behaviour, may support children in regulating attention. In their article, Waller and her colleagues [38], highlight the importance of early age in EFs' development. According to a multitude of studies, children who present difficulties in EFs during preschool age experience more psychological and physical health issues in their adult life, as well as lower rates of school readiness and later academic progress, compared to children who presented less difficulties or none at all [39,40]. The importance of EFs, both short-term and long-term, as well as their impact on children's development, has led a significant number of researchers to search for appropriate early intervention programmes, both for children with typical development and children with diagnosed special educational needs with/out disabilities (SEN/D). Primarily, these interventions are mainly computer-based programmes. However, in recent years, a promising correlation has been shown between physical education and psychomotor training with the improvement of EFs in children.

The present study constitutes an empirical research to examine the applicability and acceptability of POTENTIAL. This innovative psychomotor education programme aims to improve attention and listening skills in modern early years education. POTENTIAL is classroom-based and designed to be implemented by the teachers, utilising the existing equipment and resources. It consists of a set of 35 psychomotor sessions. It is based on previous educational, special education, physical education, and cognitive psychology research. The young participants will be introduced to the concept of attention and EFs holistically and playfully through familiar activities. POTENTIAL is designed for early years children for two crucial reasons: (i) the neuroplasticity of young children's brains poses an excellent opportunity for improvement through enriched experiences such as PA, and (ii) according to Stodden and his colleagues [41], in early years, when cognition is underdeveloped, children are more likely to have a false perception of their motor competence. As a result, they can participate and persist in physical activities even if they are not as skilled as they think, leading to skill acquisition. This false perception tends to weaken as the children grow older, which can inhibit their participation in said activities and improvement of both motor and EF skills. As it is evident, the purpose of POTENTIAL is twofold: (i) to help early years children strengthen their EFs through psychomotor education, and (ii) to promote the concept of physical literacy (the young participants learn the importance of movement not only as a means of improving their fitness, but also as a tool to promote their overall development and maintain their health and well-being).

### 1.2. Rationale for Conducting a Feasibility Trial

The unique and innovative nature of POTENTIAL (for more information, see Section 2.4.1) made a feasibility trial essential for several reasons. Before conducting a definitive POTENTIAL trial, it was significant to further our knowledge and understanding of the intervention and how it would exist in a real-life context in terms of feasibility and acceptability. Any intervention's effectiveness can be minimised and lose significance if the intervention is not practical for the person implementing it, is not realistic (in terms of duration, resources, requirements, and flexibility), and is not enjoyable for the participants [42]. Thus, the current feasibility trial allowed us to investigate such essential issues before exploring the effectiveness of the intervention. Additionally, the current feasibility trial enabled us to test the study design (procedures around recruitment, sampling, ran-

domisation, and assessment), giving us valuable information regarding whether a future definitive study of a similar design would be possible.

*1.3. Study Aims and Objectives*

This study was designed to explore the feasibility of a future definitive cluster randomised trial (CRT) to explore whether the psychomotor programme POTENTIAL is more effective than "education as usual" in supporting young children's attention and listening skills by enhancing their EFs. The objectives of the study were:

1.  To investigate the feasibility of POTENTIAL (implementation of the intervention and the study design).
2.  To test the suitability of the outcome measures completed by teachers, children, and their parents and carers.
3.  To explore the acceptability of POTENTIAL by teachers and young participants (acceptability of the intervention and the accompanying manual).
4.  To investigate the extent of the compliance and fidelity of the intervention delivery.
5.  To locate and identify facilitating and hindering parameters of the implementation of POTENTIAL (recruitment, sampling, consent, and follow-up rates).

## 2. Materials and Methods

This section outlines the study design and methodology and a detailed presentation of the programme POTENTIAL. The study was designed and outlined on the grounds of the CONSORT 2010 [43].

*2.1. Study Design*

This study was a two-arm, cluster-randomised feasibility and acceptability trial with a parallel group design conducted in Achaia, Greece's third-largest Regional Unit (with 313.766 permanent residents). Four early years classes, each with an inclusion unit, were randomly allocated to each arm of the trial: (i) POTENTIAL (experimental condition) and (ii) education as usual (no intervention condition) with a 1:1 ratio. Both class teachers in the experimental condition administered the experimental POTENTIAL (In Greece, each class with an inclusion unit typically has two teachers: a mainstream teacher who is responsible for the typically developing children, and one special educational needs and disabilities specialist teacher who is responsible for the young children in the inclusion unit.) daily for seven weeks (total of 35 sessions) after relevant training. The children's outcomes were measured by the researcher at two time points: one week pre-intervention (baseline) and one-week post-intervention. Throughout the trial, the teachers and research team continuously communicated to provide support, feedback, and clarifications on the activities and framework. Furthermore, the teachers completed a questionnaire in the form of a diary after the completion of each session to provide feedback regarding the acceptability and feasibility of each session's activities and valuable information on the implementation. Components of Steckler and Linnan's model [44] that are pertinent to classroom-based interventions were also utilised in the diary, such as the fidelity (the extent to which the programme was implemented and the activities delivered, as described in the provided manual), the dose (the amount of invention offered, measured in minutes per day and the number of sessions per week), and the dose accessed by individuals (the extent to which the young children participated in the sessions).

*2.2. Recruitment and Sampling*

The target population was early year classes with inclusion units, their teachers, and preschool children four to six years of age. Due to the nature of the study, no formal a priori power calculation took place [45]. Thus, the recruitment targets (classes and young children) were set upon what was considered necessary and realistic to investigate POTENTIAL's implementation in real-life conditions. The aim was to include four early years classes (two for each condition), each with 25 children on average. Eight children were randomly

selected in each class to test the outcome measures. The total number of participants selected to test the outcome measures was 32. Following the study design of similar classroom-based interventions [46], to portray the range of abilities found in a typical class with an inclusion unit, a stratified sampling frame was used, dividing the sample size into three sub-groups: (i) children about whom teachers had concerns regarding their attention and listening skills, but did not have a diagnosis of SEN/D (*n* = 4 per class, *n* = 16 in total), (ii) children with diagnosed SEN/D (*n* = 1 per class, *n* = 4 in total), and (iii) typically developing children with no concerns about their attention and listening skills expressed by their teachers (*n* = 3 per class, *n* = 12 in total).

### 2.3. Randomisation and Blinding

The present pilot study was a multicentre, open, randomised, controlled study of two parallel groups. Due to the nature of the intervention (universal application by the teachers without the presence of third parties), it was not possible to make the study blind at any point, either in terms of the research team or in terms of the teachers. The randomisation of the study aimed to avoid a biased selection of participating schools by the researchers. Finally, parallel groups were used to prevent the carryover effects that would likely occur if the same students participated in both groups (experimental and control). The procedure for cluster randomisation was done using sealed and opaque envelopes, with each envelope corresponding to one class. The mentioned envelopes were sealed by an independent member of the University of Patras, who did not participate in any other way in the study. Finally, the envelopes placed in each group (experimental and control) were selected by a third independent member of the University of Patras.

### 2.4. Interventions

#### 2.4.1. Experimental Intervention: POTENTIAL

POTENTIAL is a theoretically underpinned, multi-component, psychomotor education programme that targets the foundational EFs (working memory, inhibition, and cognitive flexibility) to enhance attention and listening skills in early years children (4–6 years old). It is grounded upon the systems theory and the Facilitating Intervention Training (FIT) model [47]. The theoretical framework regarding the tasks included in POTENTIAL was based on evidence from a thorough systematic review by Diamond and Ling [42] regarding the components that constitute an intervention targeting successful EFs, as well as on a more recent review by Rouvali and Riga [10] on the relationship between motor skills and EFs. Both reviews highlight that cognitively demanding PA has been one of the most promising ways of improving EFs in near-transfer activities [10]. Thus, a significant number of POTENTIAL's activities and tasks are executive-loaded, a common component in most effective programmes [42]. These activities include group games that require children to be physically active while holding and manipulating information (working memory), inhibiting responses, and adjusting to changing rules (cognitive flexibility) (e.g., Table A1).

Further to these group games, other activities that have provided some interesting and promising results regarding the enhancement of attentional skills are yoga and mindfulness. Cohen and her colleagues [48] have reported improvement in inattention in children who followed a six-week yoga intervention programme. Similar results were pinpointed by Razza and her team [49], which introduced mindful-based yoga to preschoolers. In POTENTIAL, mindfulness is part of the relaxation section of the routine, and it is provided to the children through yoga stories, breathing exercises, and guided imagery. According to Mellenthin [50], guided imagery combines relaxation techniques with creating mental images while importing all senses, which Murdock suggests can benefit young children's visual working memory skills [51]. Its overall effectiveness has been long proven in the context of play therapy. Still, its effectiveness in classroom-based intervention has yet to be thoroughly reviewed and established.

In addition to directly training working memory skills and cognitive flexibility, POTENTIAL includes another activity reported as potentially beneficial; fantastical play [52]. This fantasy-oriented play differs significantly from standard pretend role-play, where the participants act out scenes and scenarios from everyday life (i.e., preparing a meal or visiting the doctor). During a fantastical play, the children act out scenarios that they create themselves (preparing a meal for an alien or visiting a faraway fairyland). This activity may support young children's EFs in multiple ways as they are cognitively demanding and require them to alternate between fantasy and reality while holding and manipulating information (rules/script) [53].

Systems Theory of Change

The research team (a SEN/D specialist early years teacher and a psychomotor education expert) developed the framework around which and how POTENTIAL's activities would be delivered in the classroom. Other early years teachers also provided valuable additional input and constructive feedback. The logistics behind the intervention model and the implementation were also grounded in the socio-ecological model [54]. More specifically, the relationships between the resources, activities, and desired outcomes were outlined in a visual manner [55]. This way, the research team was able to pinpoint possible multi-level factors that could impact the programme's delivery in the classroom, e.g., the timing of sessions, the availability or lack of resources, and teachers' training needs.

Consequently, following Rowe and her colleagues [55], POTENTIAL's implementation and delivery were grounded upon the theory that the activities would work in the classroom context when staff was supported with adequate training and a detailed manual. Teachers received a training session before the commencement of the intervention period, as well as continuous support and clarifications. The specifically designed POTENTIAL toolkit provided detailed session plans and accompanying materials, such as visual support and premade worksheets and materials.

The Facilitative Intervention Training [FIT] Method

The FIT method is a vital element of POTENTIAL's design. According to Rapport and his colleagues [47], the FIT method is the most effective way to improve attention and other EFs. The emphasis is on things beyond possible difficulties and deficits, but on strengthening the existing skills. A fundamental tenet of programmes based on this methodology is that sustained and quantitative improvement in the development and/or efficiency of the neural systems involved in EFs can be achieved through extensive training involving repetition, practice, and feedback. This way, knowledge can be generalised to other tasks, activities, and capabilities based on the same neural systems [56]. The FIT method is also in complete agreement with the neurocognitive model of attention proposed by Posner 30 years ago. According to Posner [35], the neural network of executive attention includes the anterior cingulate cortex and forebrain regions [35,57]. The attentional functions in this neural system overlap to some extent with the EFs of working memory, cognitive flexibility, and inhibitory control [58].

Task Progression

Research has pinpointed the significance of adaptive training, i.e., task difficulty that adapts to each participant's unique level of performance [56]. As POTENTIAL is designed to be a whole-class intervention, a range of abilities among the participants was to be expected. Suggested difficulty variations at the end of each activity card enhanced all activities and games. Due to the novel nature of POTENTIAL (i.e., whole class, teacher-led, aimed for early years children), great emphasis was given during the training sessions for the teachers to understand the significance of identifying the unique characteristics of their group and choosing the appropriate level of difficulty. This feasibility trial also assessed the appropriateness of the proposed difficulty levels for 4–6-year-olds. Emphasis was placed on the proximal zone of development: the games needed to be neither too

easy, such as would risk causing boredom in the children, nor too difficult, such as causing discouragement or dropping out.

Dosage

The duration and intensity of intervention targeting EF are often poorly reported [42]. The duration of POTENTIAL (seven weeks) is based on the best available data regarding similar intervention programmes that have proven effective [59,60]. The interventions mentioned above were designed to be implemented three times a week. However, according to Diamond's and Ling's [42] review, the existing data indicate that gains in EFs are dependable on the amount of time practising (the more, the merrier). This view comes in complete agreement with Ericsson's theory about the vital importance of practising with an increasing level of difficulty [61–63]. Thus, POTENTIAL was designed to be implemented daily rather than a few times a week.

Structure of POTENTIAL Sessions

POTENTIAL is created in the form of a toolkit. It includes a teacher's manual, eight novel interactive stories, and a set of supporting materials. The teacher's manual is divided into two parts. The first part provides the reader with the framework of the programme, a brief overview of the vocabulary around EFs and the programme's components, and the theory that supports their possible effectiveness. It also includes the weekly structure of the programme, as well as a detailed suggested structure for each individual week (a total of seven weeks). The second part contains a total of 89 activities/games in the form of cards that are divided into three sub-categories: (i) the alerting activities/games (identified by a red frame), (ii) the organising activities/games (identified by a yellow frame), and (iii) the relaxing activities (identified by a green frame). The last sub-group (relaxing activities) is further divided into three types of relaxation: (i) yoga stories, (ii) guided imagery, and (iii) breathing exercises. According to POTENTIAL's structure, during each session, the children are invited to take part in one alerting activity, up to two organising activities (at least one), and one relaxation activity. This set structure aims to provide the children with a routine that will enable them to know what to expect, and feel calmer and more motivated to participate. All cards, also, include a visual support to assist the participation of all children in the class.

POTENTIAL followed a weekly and daily format that remained the same for the duration of the intervention (Table 1). Each week's topic and focus were introduced through an interactive story every first day of the school week (mostly Mondays unless Monday was a bank holiday or a field trip day). They were concluded every last day of the school week (unless bank holiday or field trip). Every session commenced with a discussion about the topic/focus of the week (or the interactive story of the week), followed by three activities. The first activity included cognitive-loaded PAs and games to alert the young participants. The second activity aimed to enable the children to organise themselves better and target specific EF skills. Finally, the last activity's purpose was to allow the children to relax and transition more calmly into the rest of the day.

**Table 1.** POTENTIAL's weekly structure.

| Interactive Story | Alerting Activity | Organising Activity | Relaxation | | |
|---|---|---|---|---|---|
| | | | Guided relaxation | Breathing exercises | Yoga stories |
| One per week (Monday part A and Friday Part B-Conclusion) | One per day | One or two per day (Target 15–20 min total duration) | Two times per week (Different each time) | Once a week | Two times per week (Same story twice) |

The Characters and Interactive Stories

One of the innovative features of POTENTIAL is the introduction of fictional characters to children. These characters were inspired by Reflecto [64] and a similar intervention by Volckaert and Noël [60]. In these interventions, several characters, each with a different job representing one of the EFs, were introduced to the young participants to enhance their metacognition. However, in the context of POTENTIAL, using characters serves multiple purposes and is extended.

POTENTIAL is designed as a treasure hunt. Before the seven-week intervention period commences, a fictional character named POT, the inattentive robot, 'visits' the young children and, through a fun interactive story, invites them to help it complete a treasure hunt in space to find its lost attention. This way, the young participants become heroes who have a collective goal. Throughout the intervention, POT and the young participants visit seven stars (one for each week). On each star, a fictional character representing one of the EFs gives the group a set of games and activities in the form of challenges. Should POT and the participants complete these challenges within a week, the character will provide them with further information about their next destination until the last week when they uncover the secret to POT's lost attention (see Table A2). Regarding the use of characters, Volckaert and Noel argue that it "allows activation, in a single image, of a set of mental representations already present in the child's repertoire" ([60], p. 41).

In POTENTIAL, children first meet Mr Brain, an expert in working memory difficulties. After providing the group with valuable tips on enhancing their working memory, he gives them a set of challenges. The second character, a researcher named Dr Flexy, provides the group with a set of challenges and tips targeting their cognitive flexibility. During their journey, the group also meets in order: (i) a witch (for inhibition control), (ii) a maintenance man (the Greek word 'συντηρητής' (syntiritís) is phonologically close to the term 'sustained' for 'sustained attention'), (iii) the twin detective sisters (auditory and visual selective attention), (iv) a traffic controller (self-regulation), and an event coordinator (planning and organisation). A poster with a pictogram of each week's character alongside tips is presented to the children before every session. These posters are on display in the class, and the teachers could also refer to them during the day when appropriate. For example, when the teacher gives instructions during a group activity, they could refer to Mr Brain's poster that offers tips on remembering instructions (repeat the information in your head or visualise it).

Using the treasure hunt storyline and the characters aims to make POTENTIAL an interactive and fun way of promoting metacognition in the young participants and active participation and engagement. In their review, Diamond and Ling [42] highlighted the two often overlooked elements of a successful intervention. More specifically, they argued that "unmet emotional, social, or physical needs will work to oppose any improvement in EFs from the programme". ([42], p. 41). Thus, they suggested that intervention programmes should also target these areas. The treasure hunt allows young children to work together as a team of heroes to achieve a collective goal. This sense of belonging may make them feel part of the group and enhance their self-esteem while improving their fitness. Furthermore, according to the same review, the benefits of PA on cognition may be influenced by the extent to which participation is enjoyed. Finally, the introduction of the characters is made for the first time through interactive stories, as children are more motivated by stories and tend to make better sense of and engage with the world in a more meaningful way [65].

### 2.4.2. No Intervention Control: Education as Usual

The classes in this condition did not receive any intervention during the seven-week trial and continued their daily routine according to the national guidelines for early years education.

*2.5. Outcomes*

2.5.1. Primary Outcome Measures—Feasibility and Acceptability

Table 2 presents the current study's primary outcomes regarding the procedures' feasibility and their link to the study's objectives.

**Table 2.** Provides an outline of the primary outcomes of this trial.

| Objectives | Outcome Measures |
|---|---|
| To investigate the feasibility of POTENTIAL (implementation of the intervention and the study design). | The rates of actual recruitment compared to the recruitment targets were counted in terms of the following: <br> • Number of classes (clusters) recruited <br> • Total number of children recruited for outcome measures <br> • Number and proportion of children recruited in each of the three subgroups within the stratified sampling frame |
| To test the suitability of the outcome measures completed by teachers, children, and their parents and carers. | • Number and percentage of sessions delivered in each class <br> • Teachers' and parents' comments on the questionnaires |
| To explore the acceptability of POTENTIAL by teachers and young participants (acceptability of the intervention and the accompanying manual). | • Measures of compliance and fidelity as an indication of acceptability <br> • Comments found in the teachers' diaries <br> • Comments found in the teachers' evaluation questionnaire (completed at the end of the intervention) |
| To investigate the extent of the compliance and fidelity of the intervention delivery. | • The number and percentage of standardised assessments, teachers' questionnaires, and parents' questionnaires completed at both time points (pre- and post-intervention) <br> • The researcher's report regarding the duration and difficulty of completing the assessment tools with the children |

2.5.2. Secondary Outcome Measures—Children's Executive Functions

In the existing literature, a significant variety of outcome measures and assessment tools have been used to test the effectiveness of interventions. In some cases, indirect measures, fully like cognitive tests [66], were used instead of direct ones, while in other cases, only objective measures were used [67]. In this trial, a combination of both direct and other-reported measures was utilised ($n = 8$). However, in the context of this trial, the variety of measures was not used to assess the effectiveness of the proposed intervention, but rather to examine their appropriateness in a future definitive study. Due to the large number of outcome measures per child, the assessment period for each child was split into four days (two days pre-intervention and two post-intervention). We were also cautious to alternate between verbal and non-verbal tools.

Working Memory

Verbal working memory. A variation of the word span task, which is standardised for the target population (Greek preschool children), was used to assess the verbal short-term memory (phonological loop) [68]. More specifically, this sub-scale assesses the ability to recall a short list of unrelated words. After the examiner reads the words (one per sec), the child is asked to recall them. If a child does not successfully recall all the words, the list is repeated until the child correctly recalls them, with a maximum of five attempts. Cue support is then given to enable the child to encode the words better. After a short break (5–10 min), the child is asked to recall the words with free recall and cue support. The number of words recalled during the first attempt and the number of attempts needed to recall the list is recorded. Also, every child receives one point for each word successfully recalled in each attempt (with a maximum of 25 points per child).

Visuospatial memory. A variation of the Corsi block tapping task, standardised for the target population (Greek preschool children), was used to assess the visual-spatial short-term memory [68]. The visual memory subscale assesses the ability to recall the

locations of a specific number of coloured 'tokens' that the examiner has placed in specific locations on a grid frame. The examiner places the tokens on the frame in front of the child, leaves them for a few seconds, then removes them and asks the child to put them back in their correct position. The child is given the number of tokens needed for the task plus two extras. The process is repeated until the child places them in the correct positions, with a maximum of five attempts. After a short break (5–10 min), the child is asked to place the tokens in the correct places. The number of tokens correctly placed during the first attempt and the number of attempts needed to position all the tokens correctly is recorded. Also, every child receives one point for each token successfully positioned in each (with a maximum of 25 points per child). Also, a point is deducted from each child for every extra token placed (if they put six tokens instead of the requested five, then one point is deducted).

### Inhibition

Cat-dog-fish. As Volckaert and Noël [60] described, the cat-dog-fish task is inspired by the Day/Night test [69] and assesses inhibitory control. This task includes two conditions: the control condition and the inhibition condition. In the former, a card with 24 images (cats, dogs, and fishes) is shown to the child, who has to name them as quickly and accurately as possible. In the latter condition (inhibition), the examiner informs the child that they are going to be 'silly' and that from that point onwards, 'cats' are called 'dogs', 'dogs' are called 'cats', and 'fishes' are called 'fishes'. The child is invited to follow the new condition and give the 'silly' animal names for the animals on the second card as quickly and accurately as possible. The reliability of this test measured by Cronbach's alpha is excellent for the inhibition condition (0.92) [60]. The number of correct responses is scored.

### Attention

Cats [70]. This assessment constitutes a cancellation task measuring selective visual attention. The child is requested to cancel as many cats as possible without errors and without paying attention to distractors. The maximum duration is set to 180 s. The number of correct responses minus errors is recorded. The internal consistency is good (0.71) and the test-retest stability correlation is 0.62 [60]. This specific task was selected above the rest of the attention-related tasks as the aspect of attention most aligned with working memory is selective attention [42].

### Cognitive Flexibility

The Dimensional Change Card Sort (DCCS) is standardised in children from three to seven years old and is a well-known measure of cognitive flexibility and a simplified version of the Wisconsin Card Sorting Test (WCST) [71]. In the standard version, the child is presented with several cards and asked to initially sort them according to one dimension (e.g., colour) and then according to a different dimension (e.g., shape). This criterion requires the child to shift their attention between phases (pre- and post-switch). A third sorting dimension is added in the advanced version of the task (used in this trial) (e.g., border vs. non-border). This version requires children to shift the dimensional focus repeatedly across phases (i.e., from card to card). The scoring of this task was based on Zelazo [71].

### Self-Regulation

Head–Toes–Knees–Shoulders-Revised [72]. This task is standardised in children from three to seven years old and was initially composed of three parts, but a fourth part has been included in the revised version, which targets verbal inhibition. In the first part, the child is asked to say 'toes' when the examiner says 'head' and vice versa. In the second part, the child is asked to 'touch' their head when the examiner says 'touch your feet' and to touch their feet when the examiner says 'touch your head'. In the third part, two more body parts are added. In addition to the previous instructions, the child is now requested

to touch their knees when the examiner says 'touch your shoulders', and vice versa. In the final part, the rules are changed, and the child is now requested to touch their knees when the examiner says 'touch your head' and to touch their shoulders when the examiner says 'touch your feet' (and vice versa). This final part is administered only if the child correctly answers at least four out of ten items in the second part. Eight practice items initiate each part to ensure the child understands the rule. The number of correct responses is calculated. This task is standardised only in typically developing children. Thus, children in the current study with a diagnosed SEN/D did not participate in this task.

Parents' and Teachers' Questionnaires

To our knowledge, no questionnaire or scale is standardised for parents and teachers of early years children in Greece. Thus, in the context of this trial, five-point Likert scale questionnaires for parents, carers, and teachers were created, including statements for working memory, inhibition, cognitive flexibility, and attention. A developmental psychologist supervised the questionnaires' development. The questionnaire consisted of 32 items and the value for Cronbach's Alpha for the survey was $\alpha = 0.908$. These questionnaires aimed to detect changes between baseline and post-intervention for each child and not compare the young participants to the norm or the young participants with each other. Parents' and teachers' questionnaires included the same statements with minor differences (the word 'school' in the teachers' questionnaire was replaced with 'home' in the parents' one)—this similarity between the two questionnaires allowed for better comparison.

*2.6. Data Analysis*

Concerning the primary outcomes, as presented in Table 2 (Section 2.5.1), the recruitment rates were analysed using descriptive methods (means and percentages). The qualitative data collected from the teachers' diaries and the two meetings with the research team were recorded and analysed using the approach by Braun and Clarke [73]. More specifically, all diaries and notes from the meetings were read, re-read, and coded with semantic codes, utilising each person's wording. Then, the emerging list of codes was organised into themes. These themes were considered and discussed multiple times before finalising them, and all codes were sorted into the appropriate theme.

Regarding the secondary outcomes, all data collected from the standardised assessments were not further processed and analysed regarding POTENTIAL's effectiveness as this trial constituted a pilot study, and aimed to only assess the appropriateness and acceptability of the programme and the outcome measures with the intention of conducting a larger scale future study to explore its effectiveness.

*2.7. Ethical Issues*

The current trial concerned a low-risk study as the activities in which the students and teachers participated did not differ significantly from the daily educational practice in Greek early years settings. The key ethical issues concern the participation of very young children and children with diagnosed SEN/D. For this reason, the primary criterion for students' inclusion was written consent obtained from parents or carers. Being fully aware of the risk while respecting the international convention on children's rights and the right of each child to express their wishes, POTENTIAL was designed to give the young participants the freedom to control their degree of participation. The procedures for obtaining written consent from parents and carers were designed based on the directives drawn up by the European Parliament and the Council on the protection of natural persons against the processing of personal data (E.U. 2016/679) and all subsequent corrigenda. A detailed protocol, including the study design, was submitted and approved by both the Committee of Ethics of the University of Patras, Greece (protocol code 12825), and the Assembly of the Department of Educational Sciences and Early Childhood Education of the University of Patras, Greece (protocol code 56649).

### 3. Results

This section outlines the results regarding the research objectives of the current study: (i) the feasibility of POTENTIAL (implementation of the intervention and the study design), (ii) the compliance and fidelity of the intervention delivery, (iii) the acceptability of POTENTIAL by teachers and young participants (acceptability of the intervention and the accompanying manual), and (iv) the suitability of the outcome measures completed by teachers, children, and their parents and carers.

*3.1. Recruitment and Sampling*

3.1.1. Recruitment Rates

The CONSORT flow chart [43] in Figure 1 below presents the recruitment procedure and rates. The recruitment targets were met regarding the number of classes (*n* = 4) and children (*n* = 32). A small number of preschools with inclusion units (*n* = 10) were invited to a presentation meeting regarding POTENTIAL, the implementation requirements of the programme, and the inclusion and exclusion criteria. After implementing the exclusion criteria, four classes responded to our invitation to participate in the study. Since there were two schools with two classes each, each school cluster was randomly allocated to the experimental or control condition. In terms of retention, no schools or individual participants dropped out of the study.

3.1.2. Sampling

Table 3 provides details of the numbers and characteristics of the classes (clusters) and the individual participants recruited to the study compared to the recruitment targets. This table shows that the study's protocol's targets were met. However, an interesting finding from the procedure was that many classes that otherwise met the criteria were excluded due to not being a class with an inclusion unit. The overall rate of parental consent (72%) was reasonable, and the desired proportion of children in all sub-groups was achieved. The only issue that arose from the process was that, in one case, a child was scheduled to be assessed for a variety of difficulties soon after the completion of the study, so the teacher did not feel confident about whether they should be put in the sub-group of difficulties or the diagnosis one. Thus, it was agreed by both parties that this specific child would be excluded from the data collection. All children, parents, and teachers completed the post-intervention assessments, resulting in no loss to follow-up (0%).

Participants

Thirty-two children (16 boys and 16 girls) were recruited in preschool classes in Greece's capital city of Achaia, Patras. Parents were sent a detailed information letter and a consent form in their child's school bag. After collecting the signed consent agreements from parents, children were seen at school for the assessment. The children's ages range from 46 to 76 months (mean = 58.9 months and SD = 8.68 months).

**Table 3.** Participants' targets vs. recruitment.

| Participants | Recruitment Targets | Numbers Recruited | Characteristics |
|---|---|---|---|
| Classes | *n* = 4 | *n* = 4 | Early years classes with an Inclusion Unit in a small part of Achaia, the third largest Regional Unit of Greece |
| Children | *n* = 32 | *n* = 42 | Gender: girls (*n* = 16, 50%); boys (*n* = 16, 50%) Age at baseline, 47 to 77 months (mean = 58.9 months) |
| | *n* = 16 | *n* = 20 | Children about whom the teachers had concerns regarding their attention and listening skills |
| | *n* = 12 | *n* = 16 | Typically developing children about whom the teachers had no concerns regarding their attention and listening skills |
| | *n* = 4 | *n* = 6 | Children with diagnosed SEN/D |

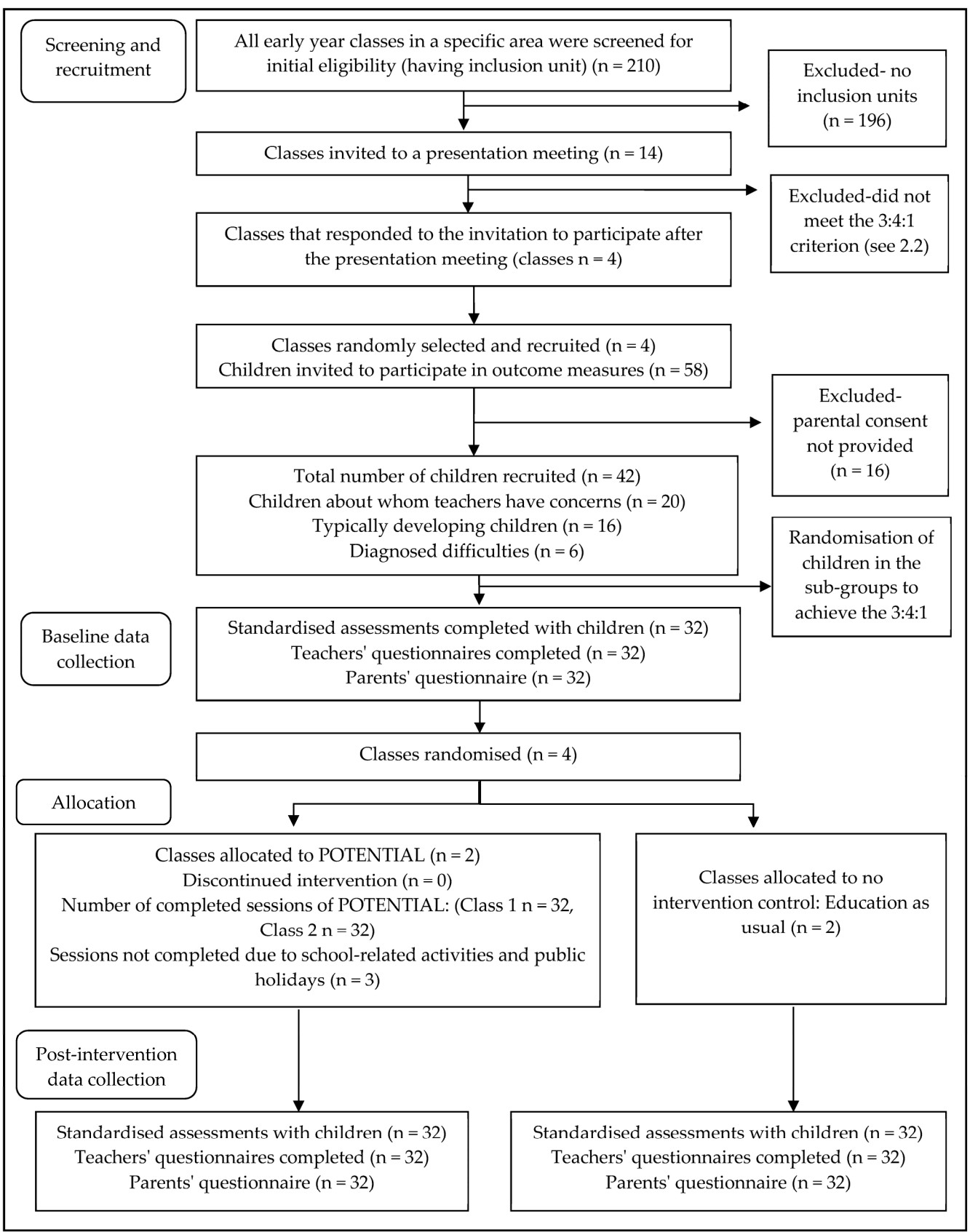

**Figure 1.** POTENTIAL cluster randomised feasibility trial flow chart (according to CONSORT, 2010) [43].

3.1.3. Compliance and Fidelity to the Intervention Delivery

The data collected through the teacher's diary indicated good compliance with the intervention delivery, as 91.4% of the sessions were completed. Regarding the sessions' duration, teachers highlighted that the activities took longer than anticipated in the first week of implementation (an average of 55 min/per session instead of the recommended 45 min). However, after consultation with the researcher and as they became more familiar with the programme's flow and structure, this issue was eliminated for the remaining six weeks. In terms of the delivery of the activities, 100% of teachers reported that the manual was straightforward to understand and follow, and that the times that they felt the need to change the wording of an activity description was minimal (less than three times among the 89 activities of the programme). The only issue that arose was that two of the eight interactive stories and two of the yoga stories included in the programme were considered lengthy by the teachers. However, this did not affect the young children's motivation and participation rates. According to one of the teachers:

> *It was the first time we incorporated a project that lasted so long in our routine, and it was so intense (daily), and not a single child reported any dissatisfaction. On the contrary, all children requested the programme daily and seemed equally excited and happy in each session from beginning to end.*

*3.2. The Acceptability of POTENTIAL*

The data collected from the teachers' diaries and the two meetings (one halfway through the study and one after its completion) with the research team indicated positive findings regarding POTENTIAL's acceptability by both the teachers and the children. Each theme is outlined in Table 4, accompanied by a quote from participants (derived from the diaries and the two meetings).

**Table 4.** Qualitative data themes identified in teachers' diaries and group meetings.

| Theme | Quotes from Teachers |
|---|---|
| All activities were acceptable. | All the activities were fun and easy to administrate. No instances did anyone report that they did not want to participate in the activities. |
| Some stories were lengthy. | The stories helped with the children's motivation. However, a few stories (two) were a little lengthier than needed. |
| The fantastical characters and interactive stories were acceptable and motivating. | Both the children and I enjoyed the stories and the characters. They were age-appropriate, and the illustration was very on point. The main character was likeable, and all the children wanted to help. |
| Need for more training before the implementation of the intervention. | Even though the continuous support from the research team was helpful, a more extensive and practical training evening would have allowed me to feel more confident during the first week. |
| The supporting material was acceptable and helpful. | The pictograms were very helpful, as they gave us smart tips about EFs in a playful way. I also liked that I could refer to them during the day (e.g., when I was giving instructions about an activity or when the children were hyperactive at lunch time). |

Theme 1: All POTENTIAL's components were acceptable.

All teachers expressed their satisfaction with the activities included in the intervention. The games and activities in the first two parts of the routine (alerting and organising) were considered appropriate, exciting, and easy to implement in the classroom environment. A finding worth mentioning was that all teachers agreed that children needed this movement break during their day, as most of them do not have access to physical education or PA. The only activity initially considered more challenging was the fantastical play (an activity repeated weekly throughout the intervention), as it was a novel activity for these settings. However, after the third week of the programme, the children showed significant

improvement that was even more apparent to the teachers after the completion of the trial. The relaxation part of the routine was reported as one of the best parts of the programme, with the young children requesting it the most throughout the day. The teachers considered a few yoga stories (*n* = 2) lengthy. Guided imagery was highlighted by all teachers as very successful, with even the most inattentive children being fully engaged, proposing to depict through drawing all the things they had imagined during the activity, which was already a suggested, but not essential, extension of the activity in the teachers' manual. In addition, all teachers reported that the supporting material was very straightforward and helpful, eliminating any possible stress from implementing a novel programme. According to them, little to no personal time was invested into the programme, and in cases that it was, the reason was that the teachers wanted to familiarise themselves with the framework further.

> *We really liked the activities in the programme. They were fun, easy to administer and age-appropriate. Even though we did not manage to work on the more difficult levels, it was nice that they existed as an option. We intend to continue using the manual throughout the rest of the school year as we saw how much our pupils and ourselves enjoyed and benefited from the activities. Unfortunately, nowadays, most children do not have the chance to move their bodies during the day or even play, so having this time during the school day feels necessary for their development and well-being.*

Theme 2: The fantastical characters and interactive stories were acceptable motivating.

It was unanimously agreed among the teachers that the fantastical characters (especially the lead character 'POT, the inattentive robot') and the accompanied interactive stories seemed to impact the motivation and participation rates of the children significantly. The characters were considered appropriately matched to each specific EF and nicely illustrated. Alongside the pictograms, they enabled the young participants to explore and endeavour to comprehend concepts (such as working memory, inhibition control, and cognitive flexibility) to the extent that the teachers reported that they did not think was possible at such a young age. The only issue that arose was that the teachers considered a few interactive stories (*n* = 2 out of 8) lengthy. In the two interactive stories, the teachers commented that a larger amount of new executive function-related vocabulary could be removed without affecting the children's motivation in the programme. However, during the programme's implementation, no child exhibited dissatisfaction or boredom due to the length of the stories.

> *The use of the treasure-hunt storyline was very motivating for the children, and the fictional characters were very well designed and matched the various EFs. Even though I was sceptical, at first, regarding using EFs terms while working with children of various abilities in an inclusion unit, I am glad that I gave it a try. The programme is well designed, and all children could participate and even understand (to various extents) the notion of attention and EFs. It needed lots of repetition and practice, but thanks to the characters and the pictograms, this came very naturally to children and me.*

Theme 3: There was a need for more training.

The programme's implementation in this trial was initiated with a training meeting with the teachers who would implement it. However, according to the teachers, longer and more interactive and thorough training would have given them more confidence and tips about implementing the programme. As mentioned earlier, during the first week, the activities took longer to complete as the teachers needed to be 100% confident regarding the implementation. However, thanks to the constant communication between the teachers and the research team, these difficulties were overcome soon after, allowing a smoother implementation for the rest of the six weeks. All the issues that emerged during this first week were documented. In a future definitive study, a training day will address these issues and provide the teachers with more opportunities for hands-on practice and interaction with the programme and its resources before its implementation.

Theme 4: The daily implementation was challenging.

A theme that emerged from the diaries and the meetings was that all teachers reported that they found the daily implementation challenging. The interesting part was that the reason for that differed from the programme. The teachers and the children were motivated and happy with the day-to-day implementation. However, due to the increasing demands placed on early years teachers over the last few years, the teachers expressed the view that, on some occasions, they felt that they did not have enough time to cover everything they needed to during the day, as most of the time after the completion of the session, the children would independently come up with ideas for extension activities. All teachers agreed that a frequency of three times a week would be more easily maintained for them, allowing them to follow up more with the children's ideas. As mentioned above, similar interventions have this frequency (three times a week).

*3.3. The Appropriateness of the Outcome Measures*

As outlined earlier, one of the study's aims was to explore the appropriateness of the outcome measures selected. The primary finding that emerged regarded the duration of the assessments with the children. More specifically, even though the research team tried to avoid this incident by splitting the data collection into two days for each child (see 2.5.2), and the completion rates of all assessment tools were 100% (both pre-and post-intervention), the process was rather time-consuming with a few children ($n = 4$) expressing tiredness and boredom in a specific assessment tool. In particular, the working memory scale (both the verbal and the visuospatial) [68] required a 10-min break, resulting in a total duration of 30 min per child which, for many children, was tiring. On the contrary, the HTKS-R [72] was unanimously accepted due to each quick (five minutes) and playful administration. In addition, the limited requirement for verbal responses in HTKS-R enabled all children (even the shier ones) to be more engaged during the assessment.

Regarding the proxy measures of the young participants' EFs, 100% of the teacher questionnaires were completed at both time points (pre- and post-intervention). Regarding the parents' questionnaires, despite a completion rate of 100%, there were a few instances ($n = 3$) where the parents spoke Greek as their second language and therefore found it challenging to complete the questionnaire on their own. However, they could complete the questionnaire with assistance from the class teacher or another family member.

## 4. Discussion

The present study aims to add to the growing body of research regarding the need for more holistic and evidence-based classroom-based approaches and programmes [10,55]. It also seeks to enhance the presence of PA in early years education to support children's overall development and well-being while promoting the concept of physical literacy [19]. To our knowledge, POTENTIAL is the first theoretically underpinned psychomotor programme in Greece to enhance attention and listening skills by improving EFs in real-life contexts. The main aim of the current trial was to explore and determine if it is feasible to conduct a CRT to evaluate whether POTENTIAL is more effective than education as usual in utilising physical education to enhance young children's attention and listening skills. The successful recruitment of both classes and children, the high completion rates, and the fact that there was no loss to follow-up indicates that the study design and processes could be scaled up for a definitive study. However, the small number of classes with an inclusion unit in the area and the emerging finding that even classes with no inclusion units could meet the criterion 3:4:1 suggests that in the definitive study, more mainstream early years classes could be invited to participate. Regarding the generalisability of the study findings to other areas in Greece, the nature of the programme (classroom-based and teacher-led) allows for POTENTIAL to 'travel' and be implemented everywhere both in Greece and abroad, should the teachers complete the appropriate training.

Regarding the second research objective, the compliance rates were good as most of the intervention sessions (95%) were delivered. This indicates that conducting a larger definitive trial is feasible to explore POTENTIAL's effectiveness. The high fidelity level (as

recorded through the teachers' diaries) also supports scaling up for a definitive trial. The only finding that emerged and called for adjusting the current study design was the need for more practical training. Moving on to the definitive trial, a more comprehensive and hands-on training session will be offered, providing the teachers with further opportunities to experiment on and interact with the resources before the implementation.

The investigation of POTENTIAL's acceptability by the teachers and young participants produced positive findings. The children and the teachers found all games in the first two sections (alerting and organising) engaging and age-appropriate. Even though, in most cases, the upper levels of difficulty were not used, the teachers commented that they appreciated the flexibility of the programme and the existence of various levels of difficulty in each activity as they intend to continue using these activities beyond the completion of the current trial. The relaxation section, alongside the stories, was the more requested part of the programme. Teachers commented that all the children (even the more inattentive ones) seemed to enjoy and benefit from the relaxation (especially the guided imagery). In most cases, they expressed the need and wished to illustrate what they had imagined. The interactive stories were accepted and reported as being both fun and age-appropriate. The EF-related vocabulary was welcomed as the stories and fictional characters enabled the children to engage with these concepts playfully. The main character, POT, the inattentive robot, became very popular among the children who expressed their wish to help it find its lost attention. Its popularity remained after the completion of the trial, with one teacher commenting that a few weeks after the trial, many children chose POT over Santa Clause as the main character in a design game. They also continued to request the stories, which were put in the school library for everyone to access. Last but not least, all teachers commented that with the completion of the treasure hunt, not only the children but themselves felt bittersweet about leaving POT, as they felt they had become a team. The children made similar comments, coming in complete agreement with Diamond and Ling's [42] theory that for an intervention to be successful, the social and emotional part of the participants should be nurtured as well. In this case, the collective goal of helping POT enabled the children to be part of a team and feel like heroes, which enhanced their self-confidence.

Concerning the appropriateness of the outcome measures (direct assessments), the descriptive statistics suggest that the following measures could be used in a full trial: the HTKS-R for self-regulation, the DCCS for cognitive flexibility, and the dog-cat-fish for inhibition control. The working memory assessment tool was time-consuming, sometimes tiring, and unmotivating for the young participants. Moving forward, the working memory tasks will be removed as they were not considered overall effective in this context and will be substituted by the HTKS-R, which has proven accurate in identifying changes in working memory [74]. Regarding the two assessments by proxy (questionnaires for parents and teachers), the high completion rate indicates they are accepted and could be used in a future definitive trial for data triangulation.

Despite not being part of the study's objectives, the comments from teachers and children (as reported by the teachers in the diary entries and meetings) highlighted another significant theme that comes in complete agreement with the existing literature; the lack of PA in children nowadays and its contrasts with the children's physiological and psychological needs [16]. It was unanimously agreed among teachers that the children's active engagement and enthusiasm towards the programme stemmed partially from their need for fun and quality opportunities for PA and movement during their day. According to one of the teachers:

> *I firmly believe that one of the reasons that all children enjoyed the programme so much (alongside its fun and engaging storyline) was that it gave them daily the opportunity to move and explore their body and its movements in new and exciting ways. As the years pass by, I notice that children in early years have less and less opportunities at school and home to engage in PA which has resulted in more and more difficulties in their overall development. It is refreshing to witness how many things children can achieve through PA and play, and how much they need and enjoy it.*

This emerging finding corroborates the literature and underlines the necessity for the Greek educational system to finally acknowledge the long-proven significance of PA and physical education, and follow the steps of other countries, such as Australia [5] and Canada [6], by adopting and promoting new guidelines about movement in early years children. Unfortunately, on the alter of standardised assessment and the narrow notion of academic success, physical education has been nearly eliminated from the early years children's lives, despite the fact that it has been and continues to be scientifically proven as a crucial part of children's overall development and short- and long-term well-being [11–15]. Moving forward, there is a significant need for more PA programmes in early years education in Greece, alongside more training for teachers, so as to learn the concept of physical literacy and its implications in young children's overall development (both short- and long-term).

*Limitations of the Present Study*

This was a small-scale trial with just two classes in each arm. In addition, the lack of a similar type of intervention in Greece (classroom-based led by teachers) prevented an active control group from being added to the study design, which would have strengthened the current findings. Finally, in the present study, two teachers led the intervention implementation in each class. In the future definitive study, it would be interesting to investigate whether the implementation would be equally straightforward in classes with only one teacher.

## 5. Conclusions

POTENTIAL constitutes an innovative, mixed-method psychomotor education programme. It aims to enhance executive functioning to improve attention and listening skills in early years inclusive education, while promoting the importance of physical education in children's lives for both their physical health and cognitive and emotional development and well-being. The current trial aimed to explore the acceptability of POTENTIAL's components and its accompanied materials and the feasibility of a future definitive study using the same protocol study design.

Regarding POTENTIAL's study design, the results indicate that a future definitive study is possible and that the recruitment process could be even more straightforward by expanding the invitation to participate to all early years schools instead of only the ones with an inclusion unit, as a significant number of mainstream classes can meet the study's criteria. The only change in a future definitive study will be reducing the number of direct assessments, as it was both time-consuming and tiring for the young participants. Concerning POTENTIAL's components and their acceptability, the teachers and the young participants accepted all activities included in the programme. The relaxation section was the most highlighted in the diaries and the meetings with the research team. The fantastical characters and the interactive stories were accepted and considered fun, age-appropriate, and a significant motivator for the young participants. The use of characters and their pictograms was also reported to be helpful for the young participants to familiarise themselves and even endeavour to copperhead the abstract concept of EFs. Its daily implementation was the only component of POTENTIAL reported as 'challenging'. However, it was pinpointed by the teachers that the day-to-day implementation was neither challenging nor tiring for the children. Instead, it was the need for more time during the school day to cover the rest of the learning objectives according to the national guidelines. Another emerging issue was the need for more interactive and hands-on practice with the materials before the programme's initiation. Despite not being in the scope of the current trial, the comments from teachers gave promising information regarding the programme's effectiveness as well.

Overall, the current study's findings highlight that with appropriate teacher training and some minor modifications, POTENTIAL could be scaled-up for a definitive study. However, a more significant finding that emerged was the children's need for more quantity and quality in the provided physical education and PA opportunities. Moving forward,

a definitive study will explore the effectiveness of POTENTIAL. Such a study will aim to prove once more that quality PA is desirable by children and can be more effective in enhancing crucial cognitive skills, such as EFs and attention, than the current approach. The policymakers should acknowledge that should we aim for the early years children to become well-rounded and developed teenagers and later adults, quality PA and education should be offered and promoted at all stages of their educational journey.

## 6. Patents

This article constitutes the first presentation of the innovative programme POTENTIAL. POTENTIAL is a multi-component, classroom-based, science-based psychomotor education intervention and prevention programme to enhance EFs to improve attention and listening skills in children attending early years education. It was designed by the SEN/D specialist teacher Aimilia Rouvali, under the supervision of the psychomotor education expert Dr Vassiliki Riga. POTENTIAL is created in the form of a toolkit. It includes a teacher's manual, eight novel interactive stories, and a set of supporting materials.

**Author Contributions:** Conceptualization, V.R. and A.R.; methodology, V.R. and A.R.; formal analysis, A.R.; investigation, A.R.; resources, A.R.; data curation, V.R.; writing—original draft preparation, A.R.; writing—review and editing, V.R.; visualization, V.R. and A.R.; supervision, V.R.; project administration, A.R. All authors have read and agreed to the published version of the manuscript.

**Funding:** This research received no external funding.

**Institutional Review Board Statement:** The study was conducted according to the guidelines of the Declaration of Helsinki and approved by the Ethics Committee of the University of Patras (protocol code 12825) and date of approval 20 December 2022).

**Informed Consent Statement:** Informed consent was obtained from all subjects involved in the study.

**Data Availability Statement:** Access to the data is available upon reasonable request by contacting Dr Vassiliki Riga at vriga@upatras.gr.

**Acknowledgments:** The research team would like to express gratitude towards all participants in the study, especially the teachers and young participants who embraced POTENTIAL and enabled us to improve it through their valuable feedback and ideas.

**Conflicts of Interest:** The authors declare no conflict of interest.

## Appendix A

**Table A1.** Structure of week 1.

| Week 1 | | |
|---|---|---|
| **Session** | **EF** | **Activity/Game** |
| 1 | Story 2- Mr Brain/Working Memory (A part) <br> Inhibition of predominant response <br> Interference of ongoing response <br> Working memory- Sustained attention- <br> Self-regulation | Storybook 2 <br> Simon says <br> Butter and honey <br> Guided imagery <br> (Let's go to space) |
| 2 | A reminder of Mr Brain <br> Attention <br> Working memory <br> Working memory- Sustained attention- <br> Self-regulation | Pictogram <br> What does your name say? <br> What is in the box <br> Yoga story <br> (Adventure in space) |

**Table A1.** *Cont.*

| Week 1 | | |
|---|---|---|
| **Session** | **EF** | **Activity/Game** |
| 3 | A reminder of Mr Brain<br>Interference of ongoing response<br>Inhibition of predominant response<br>Working memory- Sustained attention-<br>Self-regulation | Pictogram<br>Musical chairs<br>Day or night<br>Guided imagery<br>(The planet of the robots) |
| 4 | A reminder of Mr Brain<br>Attention<br>Working memory<br>Working memory- Sustained attention-<br>Self-regulation | Pictogram<br>Get the flag<br>5-s challenges<br>Yoga story<br>(Adventure in space) |
| 5 | A reminder of Mr Brain<br>Interference of ongoing response<br>Cognitive flexibility<br>Working memory- Sustained attention-<br>Self-regulation<br>Story 2- Mr Brain/Working Memory (B part) | Pictogram<br>What is the time, Mr Brain?<br>Fantastical role-play<br>Breathing exercises<br>(The shapes of relaxation)<br>Storybook 2 |

**Table A2.** The stories. (https://www.vecteezy.com/, accessed 1 April 2023 under Pro license (no attribution required)).

| Title of Story | Book Cover |
|---|---|
| POT, the inattentive robot and the pursuit of the lost attention |  |
| POT, the inattentive robot and Mr Brain |  |
| POT, the inattentive robot and Dr Flexy |  |

**Table A2.** *Cont.*

| Title of Story | Book Cover |
| --- | --- |
| POT, the inattentive robot and the Witch | 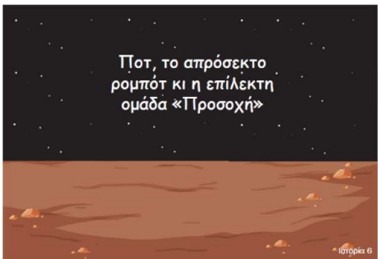 |
| POT, the inattentive robot and the Maintenance Man | 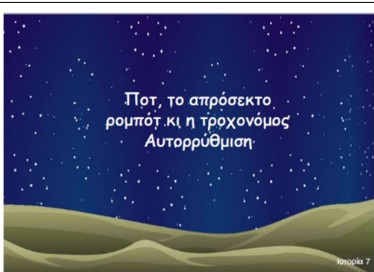 |
| POT, the inattentive robot and the detectives 'Attention' | 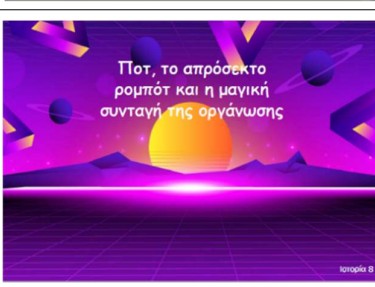 |
| POT, the inattentive robot and the Traffic Controller | |
| POT, the inattentive robot and the secret recipe of organisation | |

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
