# Peer review of "Classroom Psychomotor Education Programme to Enhance Executive Functions: A Cluster Randomised Feasibility Trial"

_2673-995X, doi:10.3390/youth3020035_

Round 1

Reviewer 1 Report

This study investigates the feasibility of a new classroom-based psychomotor education program, POTENTIAL, aimed at enhancing attention and listening skills in young children. The study addresses the growing concern of attention difficulties among children and explores a potentially promising link between physical education and executive functioning. It is conducted using a cluster randomized design, with four early years classes with an inclusion unit participating. Two classes implemented the POTENTIAL program, while the other two received no intervention. The study also discussed recruitment and sampling procedures, compliance and fidelity, acceptability of the program by teachers and children, and appropriateness of outcome measures. Overall, the study yielded positive results regarding the acceptability of the POTENTIAL program. Recruitment targets were met, and compliance and fidelity were good. The program was well received by both teachers and children, indicating its potential for wider implementation. The study's findings highlight the need for more physical education programs aimed at supporting children's executive functioning, given the increasing prevalence of attention difficulties among children. The POTENTIAL program shows promise as a potential intervention that could be scaled up in a definitive trial.

Except for the strengths, there are some minor concerns.

1. The manuscript’s structure might need to be adjusted. For example, the “6 Patents” might be moved to “1.1. Rationale for developing an intervention for executive functions” and be concise. Or might be moved to “2.4.1. Experimental intervention: POTENTIAL”.

2. Lines 386-391 “Participants” section might be moved to Line 533 “3.1.2 Sampling” section.

3. Suggest justify that the measurement instruments employed were used in the similar participants and provide validity and reliability.

Reviewer 2 Report

I congratulate the authors for the work presented, I think it is an interesting and creative way to improve the cognitive development of children through physical activity. The use of qualitative methodologies provide a lot of information to this type of study.
